# Postdocs' advice on pursuing a research career in academia: A qualitative analysis of free-text survey responses

**Suwaiba Afonja[1], Damonie G. Salmon[1], Shadelia I. Quailey[1], W. Marcus Lambert [2]***

**1** Hunter College, City University of New York, New York City, NY, United States of America, **2** Department of Epidemiology and Biostatistics, SUNY Downstate Health Sciences University, Brooklyn, NY, United States of America

* wil2009@med.cornell.edu

## Abstract

### Background

The decision of whether to pursue a tenure-track faculty position has become increasingly difficult for undergraduate, graduate, and postdoctoral trainees considering a career in research. Trainees express concerns over job availability, financial insecurity, and other perceived challenges associated with pursuing an academic position.

### Methods

To help further elucidate the benefits, challenges, and strategies for pursuing an academic career, a diverse sample of postdoctoral scholars ("postdocs") from across the United States were asked to provide advice on pursuing a research career in academia in response to an open-ended survey question. 994 responses were qualitatively analyzed using both content and thematic analyses. 177 unique codes, 20 categories, and 10 subthemes emerged from the data and were generalized into two thematic areas: *Life in Academia* and *Strategies for Success*.

### Results

On life in academia, postdoc respondents overwhelmingly agree that academia is most rewarding when you are truly passionate about scientific research and discovery. 'Passion' emerged as the most frequently cited code, referenced 189 times. Financial insecurity, work-life balance, securing grant funding, academic politics, and a competitive job market emerged as challenges of academic research. The survey respondents note that while passion and hard work are necessary, they are not always sufficient to overcome these challenges. The postdocs encourage trainees to be realistic about career expectations and to prepare broadly for career paths that align with their interests, skills, and values. Strategies recommended for perseverance include periodic self-reflection, mental health support, and carefully selecting mentors.

**Data Availability Statement:** All relevant data are within the paper and its Supporting Information files.

**Funding:** WML acknowledges support by the National Center for Advancing Translational Sciences of the National Institutes of Health under Award Number UL1TR002384. The content is solely the responsibility of the authors and does not necessarily represent the official views of the National Institutes of Health. The funders had no role in study design, data collection and analysis, decision to publish, or preparation of the manuscript.

**Competing interests:** The authors have declared that no competing interests exist.

## Conclusions

For early-career scientists along the training continuum, this advice deserves critical reflection before committing to an academic research career. For advisors and institutions, this work provides a unique perspective from postdoctoral scholars on elements of the academic training path that can be improved to increase retention, career satisfaction, and preparation for the scientific workforce.

## Introduction

Postdoctoral researchers ("postdocs") seeking research faculty positions are facing increasing challenges in their career pursuits [1–5]. The first and most dynamic is the large number of postdocs competing for a limited number of faculty positions. In the early 1970s, the number of NIH principal investigators (PIs) was equivalent to the number of biomedical postdocs and exceeded the number of graduate student researchers by more than 50% [6]. Growing from 7,000 to over 21,000 biomedical postdocs today, some argue that there is an oversupply of postdocs and overemphasis of academic tracks, leading to a hypercompetitive culture among trainees for faculty positions [7, 8]. This oversaturation may be partially attributed to insufficient preparation of graduate students for career options outside of academia [9–12], leaving postdoctoral positions as a default career step for many PhD holders [13, 14]. Others maintain that postdoc oversaturation is a misperception as postdoc positions should not be considered only as preparation for an academic job but rather as an opportunity for skill development for a multitude of fields [10, 15–20]. In addition, the total number of tenure-track faculty positions has remained relatively constant over the past few decades, and the disparity between postdoctoral appointments and available tenure-track positions has not been proportionally adjusted for [6, 7, 21]. Currently, tenure-track faculty positions only represent approximately 15% of postdoc career outcomes [22].

Second among these growing challenges is the poor sense of financial security felt by many postdoctoral researchers. Previous reports note low salaries and a long overall training length for many postdocs [22, 23]. The postdoctoral position, intended to be a temporary training period, has been increasing in length without a substantial increase in pay, with recent studies reporting postdocs completing more than one postdoc [24, 25]. McConnell et al. [22] found that postdoctoral salaries are not maintaining parity with the cost of living increases. At the time of their study (2016), postdocs reported salaries in the range of $39,000–$55,000 (median $43,750, mean $46,988) [22]. The financial sacrifices and increasing time commitment made during this training phase are compounded by a lengthy biomedical doctoral training period with a median time to degree of greater than 5 years [26].

Over the last five years, the National Institutes of Health have increased the Ruth L. Kirschstein National Research Service Award (NRSA) stipends by four to five percent on average for postdocs [27]. This adjustment is consistent with recommendations from the 2012 *Biomedical Research Workforce Working Group* report from the NIH and the 2018 *Next Generation of Biomedical and Behavioral Sciences Researchers*: *Breaking Through* report from the National Academies of Science, Engineering, and Medicine [27–29]. Many institutions are starting to follow suit, either using the NRSA levels as guidelines for setting postdoctoral salaries or setting the minimum in accordance with the Fair Labor Standards Act [30]. Historically low stipends and a sense of financial insecurity are associated with increased interest in non-academic careers [25].

Researchers who begin a postdoctoral position despite awareness of the salary support and limited availability of faculty positions may still lose interest as their training progresses [31, 32]. Some evidence suggests this is due to the incompatibility between their career preferences and the demands of the academic lifestyle [9, 33]. Unrealistic expectations or lack of knowledge about aspects of academic life such as academic freedom, administrative obligations, funding, and the time commitment may cause many to prematurely abandon this track despite already committing a significant number of years to it [33]. With regard to women and underrepresented minority (URM) postdocs, the largest exit from the academic research pipeline can be observed in the first two years of postdoctoral training [25]. An increase in transparency about life in academia and the dissemination of more information on the critical steps for securing an academic research position should increase the retention of trainees by giving those who choose to commit to this track, despite the known challenges, the opportunity to make well-informed career decisions that reflect their personal and professional values [34].

Postdoctoral researchers can provide a unique perspective on the benefits and challenges of a research career. They are in a distinctive training period that serves as the branch point for their future careers and have personally experienced many of these benefits and challenges. Our previous work identified the most influential factors for those who intend to pursue teaching or non-academic career paths (this includes teaching-intensive faculty positions, non-academic research positions such as industry, non-research but science-related positions, and non-science related positions) [25]. *Job prospects*, *financial security*, *responsibility to family*, and *mentorship from their PI* were the most cited reasons for those opting for careers outside of academia [25]. Postdocs who pursue academic research careers produced significantly more publications (9 vs. 7, p<0.001), more first-author publications (4 vs. 3, p<0.001), and have a higher first-author publication rate (0.56 vs. 0.42, p<0.001), yet, a significant portion (40%) of even the most productive postdocs opt out of pursuing an academic career [25]. Thus, a greater understanding of how postdocs perceive the path to academic research independence is warranted. In addition, strategies to overcome the challenges faced along the way are also needed.

This current study amasses the guidance and recommendations of 994 postdocs on pursuing an academic career. Our objectives are:

1. To understand how postdoctoral research trainees perceive the benefits and challenges of pursuing an academic research career;

2. To provide ways to overcome these challenges.

The advice gathered from the postdoctoral researchers in this study will help prospective trainees make more informed career decisions. Instead of only describing some of the obstacles they have faced, the postdocs provide numerous strategies and suggestions to help future and current researchers take greater control of their career outcomes. These primary accounts further elucidate many of the challenges researchers encounter when choosing career paths. Therefore, such disclosure can increase transparency about the benefits and challenges of pursuing a tenure-track faculty position and encourage a more efficient transition from training stages to careers across the scientific workforce.

## Methods

### U-MARC survey

Postdoctoral scholars ("postdocs") in the biological and biomedical sciences from across the United States were invited to complete an original survey instrument entitled U-MARC

(Understanding Motivations for Academic Research Careers) in July of 2017. The 70-item survey (1) measures views on determinants of career choice in science and (2) measures outcome expectations and self-efficacy around research careers using two original scales. The study's theoretical framework was derived from (i) Social Cognitive Career Theory which states that self-efficacy, outcome expectations, and personal goals affect career decision and (ii) Vroom's Expectancy Theory which infers that motivation is a result of how much an individual wants a reward (valence), the probability that a specific effort will lead to the expected performance (expectancy), and the belief that the performance will lead to the reward (instrumentality) [35, 36]. We used expectancy theory to build an outcome expectations instrument in the U-MARC survey, with some items taken from the Research Outcome Expectations Questions (ROEQ) [37]. For full details on the development of the U-MARC survey instrument, refer to our previous work (Lambert et al.) [25].

## Data analysis

In this study, we qualitatively coded 994 survey responses to an open-ended question from the U-MARC survey using hallmarks of both content analysis (examining patterns in text, highlighting frequency counts) and thematic analysis (interpreting themes within the data). The question states: "What advice would you give to someone thinking about an academic research career?" Two researchers were involved in the coding process, each independently deriving codes. A process of open, axial, and then selective coding was followed by generally coding and discussing major concepts, categories, and themes. A third researcher was consulted to help determine crosscutting themes and recurrent patterns, in consideration of analytic connectedness. We repeated this cycle until we achieved thematic saturation, and novel themes stopped emerging from the data. NVivo 12, a qualitative transcript software, was used to assist with the coding of the data.

Throughout the manuscript, we include transcript numbers corresponding to the survey respondents' answers so readers can differentiate between the sources of any given set of quotations. The full list of responses and derived codes are included in the S1 Table.

## Data collection and sampling method

All work was conducted under the approval of the Weill Cornell Medical College Institutional Review Board (IRB# 1612017849), and all respondents self-selected and provided consent for participation in the study. A purposeful sampling strategy where participants were recruited through postdoctoral listservs from top-ranked research universities and institutions was used instead of snowball sampling, where existing participants would have recruited other potential candidates from their networks. All survey respondents self-selected to participate in the survey based on the inclusion and exclusion criteria previously published [25]. The sample represents wide geographic (over 80 universities) and subfield diversity. The number of institutions and the percentage of respondents from each institution were published in supplementary file 2 of our previous publication [25]. We also determined the percentage of respondents from highly-ranked life science research institutions in the US based on counts of high-quality research outputs between January 1, 2017 and December 31, 2017 according to rankings from *Nature Index* (Fig 1- figure supplement 1D) [25]. The majority of respondents are from highly-ranked US institutions, but the differences in institutional ranking do not fully account for the differences in career intention [25]. It should also be noted that postdoctoral appointees at the top 100 institutions in the United States (n = 56,092) account for approximately 88% of the total number of postdocs in the country (n = 63,861) [38]. From the total sample of participants who completed the U-MARC survey (n = 1248), only respondents who identified as a

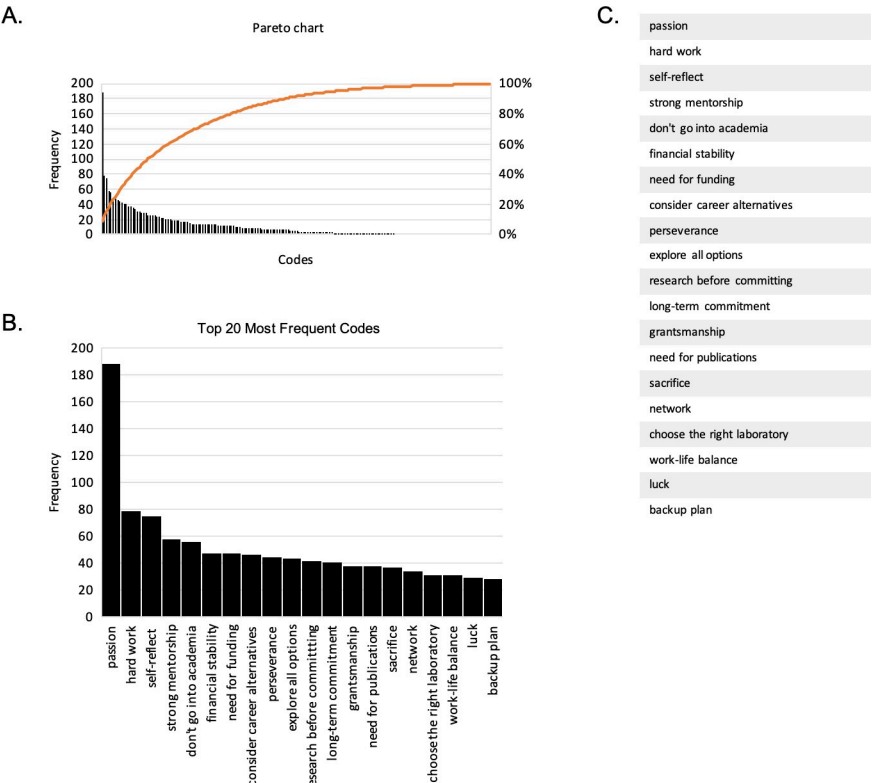

**Fig 1. Relative frequency distribution of advice on pursuing an academic research career.** (A) To estimate the prevalence of advice from postdoc respondents, the frequency of the codes was analyzed and displayed by the bars (left axis) in descending order along with its contribution to the cumulative percentage, represented by the line (right axis). (B) The top 20 most frequent codes are displayed and (C) listed by frequency.

postdoctoral scholar or research associate were included in this analysis (n = 994). The sample postdoc participant pool represents 6% of the total amount (21,781) of appointed biomedical and biological postdocs the year the survey was conducted (2017) according to the National Science Foundation. The REDCap electronic data capture tool was used to collect and manage the 70-item anonymous U-MARC survey instrument. REDCap (Research Electronic Data Capture) is a secure, web-based application designed to support data capture for research studies.

## Results

To establish a representative framework of life in academia at the postdoctoral stage of training and gather recommendations for success in this field, we asked postdoctoral candidates, "What advice would you give someone thinking about an academic research career?" The sample of 994 postdoctoral researchers included US citizens (n = 557, 56%), international fellows (n = 434, 44%), female postdocs (n = 615, 62%), male postdocs (n = 378, 38%), and underrepresented minorities (URM) postdocs (n = 174, 13%) (Table 1). URM postdocs include the racial and ethnic categories of American Indian or Alaska Native, Black or African American, Native Hawaiian or other Pacific Islander, and/or Hispanic or Latino. The participants responded similarly across gender, race/ethnicity, and national status. The average time reported to earn a PhD in the biological or biomedical sciences for the postdoctoral respondents was 4.6 years, with an additional 2.7 years spent in postdoctoral training. The average

**Table 1. Demographic information of survey participants.**

|  | N = 994 |  |
|---|---|---|
| Female | 615 | 62% |
| Male | 378 | 38% |
| U.S. Citizen | 557 | 56% |
| International | 434 | 44% |
| Underrepresented minority | 133 | 13% |
| Average time to PhD | 4.6 |  |
| Average time post-PhD | 2.7 |  |

time to PhD recorded here is lower than that of the national average due to the inclusion of the international students' shorter doctoral training lengths.

With the information gathered from the 994 responses, 177 codes and 20 categories emerged. To estimate the prevalence of advice across postdoc respondents, we determined the frequency by which each code was cited (Fig 1A). 'Passion' emerged as the most frequently cited code, referenced 189 times across 994 responses, more than twice the amount of other codes (Fig 1B). The top 20 most frequently cited codes included: hard work, self-reflect, strong mentorship, don't go into academia, financial stability, need for funding, consider career alternatives, perseverance, explore all options, research before committing, long-term commitment, grantsmanship, need for publications, sacrifice, network, choose the right laboratory, work-life balance, luck, and backup plan (Fig 1C). In the following text, we summarize the postdoc respondents' advice across two major themes: **Life in Academia** and **Strategies for Success**.

## Life in academia

**Academia is a lifestyle.** According to the surveyed postdocs, a career in academia is not merely an occupation; it is a lifestyle (Table 2). The postdocs express that the time and

**Table 2. Categories which emerged from the codes: Life in Academia.**

| Category | Codes (modified) | |
|---|---|---|
| Academic life | expectation to publish | "academia is not a meritocracy" |
|  | grantsmanship is necessary | opportunity to teach |
|  | academic freedom is an incentive | mentoring students |
|  | administrative obligations |  |
| Challenges | hard work | difficult relationships |
|  | job market is competitive | limited opportunities for advancement |
|  | academic politics | overworked |
|  | long hours | burn out |
|  | lack of recognition for work |  |
| Financial security | lack of financial stability | need for more funding |
|  | underpaid |  |
| Understand the risk | "luck is needed for success" | assess cost-benefit ratio |
|  | success is not guaranteed | be ready for setbacks |
| Wellness | mental health support | take time off |
|  | evaluate your self-worth | find a hobby |
| Work-life balance | demanding workload | parenthood |
|  | familial obligations |  |

commitment required for success in academia often shapes a researcher's personal, professional, and social life:

> . . .anyone who pursues a research career has to be [prepared to] work in a high-pressure environment that consumes your life. There is really no separation between your research and life. You must be prepared to sacrifice holidays and special occasions with your family. (379)

So, the postdocs advise performing a self-assessment of your values and priorities to determine whether this career can satisfy your personal and professional needs:

> Consider what balance of work and personal time is acceptable and will make you happy. Those who reliably and regularly receive high impact publications and large grants tend to spend a majority of their time in lab writing, and have less free time outside [of the] lab. (157)

> Set the professional goals you are willing to achieve without endangering your personal life. (197)

Respondents describe the challenges they face as postdoctoral researchers to also include long hours, a demanding workload, unanticipated setbacks, a competitive funding and research climate, and delayed gratification, consistent with other studies [5]:

> Academic research is great if you enjoy working hard, tolerate frustration, and accept that sometimes you need to work for years before seeing results. It's not that great if making money is important for you, if you need a lot of free time for your non-work life, and if you need frequent reinforcement and sense of success. (17)

The trade-offs to these challenges include scientific creativity, problem solving, academic freedom, and travel. Academic freedom was cited as one of the main benefits of pursuing an academic research career:

> If making discoveries on a day to day basis, big or small, is what you crave for then [this] is the career. If you have the patience to go through failure or unexpected results to find something new and interpret it, be criticized without giving up and then prove your point then this [is] the career. If you are curious enough to get to the truth, no matter what but at your own pace then this is the career. (427)

> Be prepared for a lot of freedom in thought and ability to pursue academic interests with a great deal of effort in lab management, grant writing, and grant management. (75)

Many also note that luck (or probability) plays a significant role in your success in experiments, publications, funding, and job opportunities. This perception has been previously linked to levels of outcome expectations among postdocs, i.e., whether their hard work leads to high performance [27]:

> You can put in 80 hours a week, but unless you get lucky, you will not be able to publish in multiple high impact journals in order to attain a position in Academia. The hyper competitive climate in science is extremely discouraging to intelligent and innovative minds. . . (951)

However, several caution against allowing your research success to define you:

*Do not base your sense of self-worth on having an academic position. Give it your best shot, but there are many viable pathways out there and you shouldn't feel that being an academic is the only viable option. (186)*

Overall, the postdocs describe life in academia as one characterized by significant time demands for conducting research and fulfilling other responsibilities. They note the importance of establishing a healthy work-life balance and advise prospective researchers to determine whether their fields of interest can meet their personal and professional expectations.

**Postdocs feel underpaid and undervalued.** Several survey respondents indicate that compensation at the postdoctoral level is unsatisfactory:

*Relative to their educational attainment and training, postdocs are very poorly paid, and there is little or no job security. (199)*

*. . .academic institutions arbitrarily devalue your contributions both financially and through the game of non-promotion. (835)*

Many call prospective candidates to be aware of these financial challenges:

*Can you live with a salary of 20k-30k as a graduate student until you are 30? Are you ready to start a family with a salary of 50k as a postdoc when your friends are making 70k-80k without a PhD? (979)*

However, the postdocs indicate that passion should be the primary motivation for those interested in pursuing an academic career, not financial gain:

*If you care about money, don't come. This field is sustained only by passion now. (448)*

*Science should be your number one passion. It must be strong enough to overlook the many long hours and the fact that you're spending your peak earning potential years in a stressful, low-pay, unstable 'training' position. (736)*

Though several postdocs feel many institutions fail to provide the level of professional development, career placement, and employee benefits to postdocs as given to students or faculty [39, 40]. These limited job prospects lead to feelings of not being appreciated in postdoc positions [41]:

*You will be mentally run-down, under-paid, under-appreciated, and in the end make less than you're worth with fewer benefits. (732)*

*I don't want my kids to go into research; I want them to do something where their work and knowledge is actually appreciated. And where this appreciation is reflected in the salary. (693)*

So, many highly recommend being proactive about choosing the best work environment to complete your postdoctoral training:

*. . .environment matters. Being at a supportive institution with excellent mentorship and opportunities in your field of choice will be important for success. (777)*

*Have a strong mentor and support system at your institution because your institutional resources will be heavily counted toward successfully obtaining a grant. (367)*

*Make sure to choose a lab that publishes regularly and a mentor who will actually act as a mentor and not just get science out of you. (594)*

The postdoctoral researchers find that the state of funding in academia has made it difficult to not only support their research but also their personal cost of living. However, they find that if you are passionate about your work and can find a supportive institution with strong mentorship, those sacrifices can be reduced and are ultimately worth the effort.

## Strategies for Success

**Prepare for multiple career paths.** Some postdocs do not recommend pursuing an academic research career at this time. Despite hard work and considerable sacrifice, the probability of obtaining a tenure-track faculty position and financial insecurity were cited as the main deterrents for pursuing an academic research career:

*Do something else. I am a successful young research scientist, but I would not advise a student to pursue a career in academic research. The balance between the effort put in, and the benefits gained is completely skewed. Dedication and hard work are no guarantee of success in terms [of] publications. Very often, early career decisions as to the lab you apply to, to do your PhD training, have an inordinate influence in your career. Also, mentors have an inordinate influence in their [students'] happiness and success. . . (199)*

*I just started to apply for tenure-track openings and have been told by dept. chairs that I need to have funding (K-award) in hand to be seriously considered for a position. I have 30+ publications and received my own funding since I was a graduate student (NIH F31 & F32 plus over $120K in supplemental funding). My K-award is currently under revision and feel my future is currently 100% dependent on my whether I get a K regardless of what is on my CV or my past accomplishments. (238)*

In consideration of these challenges, several postdocs indicate that there are many other viable career opportunities for researchers. They also suggest that academic research careers should be framed as a part of a myriad of successful post-PhD careers, rather than an alternative to those unsuccessful at achieving a faculty position:

*Please, don't consider [an academic research career] as the only respectable career for a scientist. Keep your options open. (474)*

*I think the PhD is still a worthy goal. But I've come to realize the faculty position isn't the be all and end all of the process. There are many other useful and creative and important ways to use a [PhD] degree. I think new generations are coming around to that: they embrace alternative careers and they are being trained for them in better ways as universities accept the fact that most of us don't attain the faculty job. . . Another thing is, if you get the [PhD] and you want to leave academia, make a plan for that career- don't get sucked into a postdoc when you [don't] want to become a PI. (390)*

*If you can succeed in basic research, you can be more successful in industrial, financial or other [fields]. (446)*

Some of the postdocs recommend exploring multiple career options immediately, while others encourage keeping an open mind about multiple career paths in case the academic track becomes less viable (Table 3). Pursuing a career in industry was the most cited career

**Table 3. Categories which emerged from the codes: Career planning.**

| Category | Codes (modified) | |
|---|---|---|
| Career planning | thoroughly evaluate career landscape before committing | gain research exposure early |
| | carefully assess laboratory environment before committing | conduct informational interviews |
| | have a backup plan | consider location |
| | be realistic about expectations | find your research niche |
| | be open-minded | revise career plan regularly |
| | talk to people at various stages of career | evaluate whether PhD is necessary |
| Non-academic careers | explore all career options | Master of Business Administration (MBA) |
| | industry | government |
| | translational research | entrepreneurship* |
| | finance | patent law* |
| | non-bench careers | consulting* |
| | medical school | science communication* |
| | science policy | science and medical writing* |
| Commitment | long-term commitment | long-term gratification |
| | requires sacrifice | be open to developing new skills |
| | persevere through failure and rejection | take advantage of opportunities |
| | enjoy the journey | don't remain on career track if odds are not favorable |
| | be open to different research areas | |
| Self-reflection | know your values | know your limitations |
| | establish projected time frame | evaluate strengths and weaknesses |
| | consider which aspects of research you like | imposter syndrome |
| | explore your skills and abilities | |
| Negative sentiments toward academia | "don't go into academia" | "boycott field" |
| | "academia is a broken system" | |

*added by authors to present a comprehensive view of non-academic careers.

choice outside of academia by the postdoc respondents because of its competitive salaries, structured career advancement, better work-life balance, and the opportunity to continue scientific research or make contributions in other capacities.

**Reflect on your motivation.** Passion, coupled with resilience, was cited as the primary driving force for pursuing an academic research career:

> Be sure you are passionate about your field of [study] and are pursuing the career for the right reasons, because it is not an easy career path. It allows for creativity, flexibility, and joys of learning and discovery, but is challenging in terms of funding, navigating bureaucracy and politics, administrative obligations, etc. (101)

The postdocs assert that researchers should not be driven by extrinsic factors such as wealth or fame:

> People that are successful in the academic field are not in it for the money or fame. Chances are you won't become famous or rich, but you do have the potential to help countless amounts of people, and if you are passionate about research and helping others, you shouldn't let the troubles of funding or the woes of others deter you from your goals. (415)

*Never forget that the goal of biomedical research is to eventually find targets for human health issues that will hopefully help eradicate disease. . . and that research does not occur in a vacuum and requires passion, leadership, hard-work, and collaboration. (635)*

Many of the respondents described the absence of passion as one of the leading causes for abandoning the pursuit of an academic research career. They maintain that love for the sciences and the impact of academic research is what makes the sacrifices worthwhile.

**Assess readiness.** To supplement this advice, the postdocs define the qualities of a good scientist:

*. . .true scientists care very little about money or taking the easy route. They are just intellectually curious and looking for answers to their questions. (77)*

*Make sure you really enjoy coming up with your own hypotheses, have the knowledge to assess their novelty, and the writing skills to get them funded. (613)*

*[Researchers'] daily tasks rely on numerous skills like writing communication, teamwork, problem solving, planning, learning, self-criticism, etc. (730)*

According to the respondents, good scientists are passionate researchers who conduct thorough scientific investigations, demonstrate resilience in the face of difficulty, and are disciplined leaders in their fields. They are confident, excellent collaborators, and comfortable with failure and uncertainty. Their ability to take criticism and recover from setbacks allows them to overcome rejection and persevere through the many challenges present on this track:

*Doing good science is slow and hard, and there are many times in research that it is easy to get discouraged—make sure you identify a way to reignite your passion for research so that you can overcome those times of frustration—we need more people and more diverse ideas in this profession, not fewer. (210)*

Based on these qualities, the postdocs call prospective researchers to assess their strengths and weaknesses to determine if they are well-suited for this profession:

*Do you truly enjoy research and the responsibilities (i.e. writing papers/grants) that come with those responsibilities? Are you competitive and confident in your ability to do science? Are you able to compartmentalize when research fails and not place blame on yourself? (206)*

*There is always someone smarter out there, but you can control how hard you work. Research, like many things in life, is a battle of attrition. Work hard in the lab, write a lot. . . .like more than you think you should, read often, collaborate with others, keep growing. And most importantly, don't let a paper or grant rejection define you. Your worth is derived from the things in your control, not the things out of your control. (798)*

The postdocs recommend performing an honest self-assessment of your motivations and abilities to determine whether you have the drive and character necessary to maximize your chances at successfully obtaining a tenure-track faculty position.

**Be strategic.** The postdocs emphasize the importance of being diligent and methodical about career development:

*Pursuing science for the sake of exploring how the world around us works, and planning an academic research career are two very different endeavors, which require many mutually*

**Table 4. Categories which emerged from the codes: Strategies for Success.**

| Category | Codes (modified) | |
|---|---|---|
| Passion | "field is sustained only by passion" | money (wealth) should not be primary motivation |
| | "if this is your dream, the sacrifices are worth it" | |
| Qualities of a good scientist | patient | intellectually curious |
| | confident | "thick-skinned" |
| | open-minded | disciplined |
| | resilient | strong leader |
| | creative and innovative | has integrity |
| Research skills | strengthen research skills | "commit to doing good science" |
| | strengthen writing skills | |
| Strategy | be strategic about decisions | be vigilant about opportunities |
| | be comfortable with uncertainty | have a holistic approach to advancement |
| | explore current literature | give presentations |
| | attend seminars | start simple |
| | be competitive | have multiple projects |
| | be prepared to self-learn | choose a field in demand |
| | choose the right graduate school | have specific research focus |
| | focus on the positives | utilize your advantages |
| Laboratory | choose the right laboratory | choose a famous laboratory |
| | find a well-established PI | evaluate laboratory publication record |
| Mentorship | seek strong mentorship | learn to ask for help |
| | evaluate mentorship compatibility | |
| Publications | "publications are the currency of research" | don't need to publish in top journals |
| | publish in high impact journals | |
| Network | seek collaborations | build connections |
| | find supportive community | develop communication skills |
| Transferable skills | understand skills valued outside of academia | develop skills for non-bench careers |
| | seek cross-training in areas beyond research focus | pursue managerial training |

*exclusive skills. Working towards a career in science must include careful choice of mentors, labs and projects in graduate school and post-doc. (714)*

They also advise strengthening your research skills, staying informed about current research, and finding a supportive community to grow and develop in:

*Learn how to critically read data and develop independent ideas and experiments. Work hard both at the bench and at understanding and staying current in the literature. Learn to ask for help and take criticism. Build your professional network to include [scientists] of various backgrounds and expertise. Meet and discuss your science as [frequently] as possible with these colleagues. (568)*

The following sections expand on these strategies for success (Table 4).

**Choose the right laboratory.** The postdocs stress that your choice of laboratory is one of the most critical steps for career advancement:

*Find a lab that has [a] track record of postdocs transitioning to professorships. Your postdoc boss has the most influence on your own independent academic career. [It's] all about mentorship. (279)*

*Do intensive research, not only about the area you're interested in, but the environment/ morale of the lab itself before taking a job in a lab. (175)*

The postdocs recommend carefully assessing all aspects of the laboratory environment before committing to a mentor or laboratory. This might involve conducting informational interviews to gain insight into your potential work environment, the laboratory's publication record, and your prospective PI's expectations of you. Due to your PI's influence on your future, many advise assessing mentorship compatibility before committing to a laboratory to ensure that your values and expectations are compatible.

**Choose mentors wisely.** The postdocs strongly emphasize the importance of strong mentorship on career growth:

*Find a mentor who has the time and passion to see [you] grow and is willing to explain what the different career trajectories are, how you go about finding them, and wants to assist you in that journey. You need to find a mentor who values your potential as a future scientist and doesn't just view you as cheap labor. (282)*

*The greatest component to my success thus far has been asking the right people for help with grants, experiments, and other challenges. Without the support of my community it would be difficult to push forward research initiatives and secure funding for them. (31)*

The postdocs share that insufficient support from PIs may leave researchers underdeveloped in some critical academic skills. So, they highly recommend seeking out multiple mentors at various stages of their careers because these professionals offer invaluable perspectives, skill sets, advice, and resources.

**Publication strategies to consider.** Although there is debate about the significance of journal impact factor, there is certainty among the postdocs about the need for publications to become established in the academic community:

*[Academia is] not a meritocracy. Either you have to publish a lot, or publish in high impact journals only. (515)*

*Papers are the currency of research, so it is crucial to publish extensively in reputed journals. In the job market, being able to market your 'brand' of science—topics, approach, methods, [etc.] becomes an important ingredient to success. (714)*

Many recommend strengthening your research and writing skills to maximize grant and publication success:

*Develop your brainstorming/project design skills. The opportunities to do that as a postdoc may be slim, but it is essential to being able to write your own grants and secure independent funding. (172)*

*I would recommend that they try to be as independent as they can in their research ideas, strategies and grant writing (from Grad school and on). If they are successful at each of these, and their ideas are well perceived by their field, then they likely have a shot. It is important to think about what you bring to the field and what you want to teach others through your research and mentoring. (405)*

They also advise assessing the publication record of the laboratory you are considering before committing to it:

*For grad school and postdoc [choose] labs where the PI is established in his field because this is the way to get published in high ranking journals and to get your grants approved, without which you cannot develop a long-lasting career in Academia. (312)*

*Get the training from the lab [with a] history of publishing top journals. Name of the school is not as important as the quality of the paper your candidate lab publishes. (330)*

Some suggest working on various projects to enhance your academic record:

*. . .do not only work on one project, but several, if you work on a risky project, get a second safer project to ensure you get regular publications. . . (25)*

*Having a diverse portfolio helps, rather than one big project, one big book, or one big paper in the pipeline. I built my pipeline very slowly, and that gave me stress. I do think there is a need to step outside the publish or perish game, and craft one's own rules of engagement if possible. One way to do this is possibly doing work meaningful to oneself rather than looking at where the funding comes from, or what everyone else is doing. (891)*

The postdocs highlight the importance of publishing because it gives your research exposure, builds your credibility within the scientific community, and increases your competitiveness for an academic research position.

**Network.** The postdocs find networking to be an essential component of success in academia:

*[Networking] with peers and labs you're interested in and offering your help or services or forming collaborations provides vital long-lasting and fruitful connections. [It's] often who you know that counts in terms of publications. [Get] involved with excellent research labs, learn from them and become a vital part of the team. (518)*

*Go outside your comfort zone of the lab, engage in science outreach, network, interact! Science is more than working hard at the bench, and I wish somebody had told me that earlier in my career. (685)*

The postdocs recommend attending conferences, building connections, and collaborating with others because those are important ways to develop relationships with scientists throughout the community, learn from their expertise, find mentors, and share your own research and ideas.

Overall, the postdocs' provided these recommendations to help prospective researchers make more informed decisions about their research career pursuits:

*In order to achieve a successful career in academic research, one needs to understand early enough that it is more than a job in science—it a permanent dedication to scientific topics, with a lot of workload for a single person. Persistence and patience are essential traits to succeed. (469)*

## Discussion

In this manuscript, we present the advice of 994 postdocs on pursuing an academic research career. The postdocs' responses were analyzed to derive codes that encapsulated the major concepts being discussed. We found 177 distinct codes in 20 categories across 10 subthemes

and two broad themes. In the first theme, *Life in Academia*, postdocs detail a picture of academic life from the point of view of the trainee, not often captured in the literature. According to several studies, the postdoctoral experience in the United States has not been captured comprehensively in more than a decade [7, 22]. This scarcity of data negatively impacts postdoc career outcomes and the overall vitality of the scientific workforce.

In this study, postdocs highlight both the flexibility and challenges of academic life: "It allows for creativity, flexibility, and joys of learning and discovery, but is challenging in terms of funding, navigating bureaucracy and politics, administrative obligations, etc." They also report that this track demands a significant commitment to seeking funding and publishing research, typically exceeding normal work hours. Thus, for many postdocs becoming a faculty member would fulfill their passion for research and discovery, but success requires managing the constant tension between work demands and their personal lives. Before committing to this career path, one has to decide if the lifestyle and its challenges are worth the reward.

One major challenge that the postdocs frequently referenced in this study was financial insecurity. We suspect that not everyone considers the financial impact of years of training until they are immersed in it. In this study, we find postdocs' frustrations over aspects of financial support. Some postdocs advised not pursuing this path unless you come from a wealthy family. Others were very specific about the personal cost of this education and training: "...I'm in my mid 30's and have worked 60+ hours a week for 10+ years for essentially minimum wage in hopes of getting an academic position." Such openness is needed to help future trainees have a clearer understanding of the challenges in the field and help funding agencies understand how to improve access for all trainees. Choosing to pursue an academic career should not be dependent upon how long one can sacrifice financial stability and security. Postdocs should not be driven away from academic paths because they do not have the financial means to complete a postdoc.

The increasing length of this poorly funded training period with no guarantee of success causes financial instability and job insecurity [22]. Moreover, the increasing age for establishing scientific independence also translates into why there is a limited availability of faculty positions. Zimmerman shares that the reported average age for scientists to secure their first NIH grant is 42 [7]. This data along with the fact that the number of tenure-track faculty positions has remained consistent over the past few decades may in part account for the shortage of available positions as more senior researchers maintain their tenure to reap the benefits of the personal, professional, and financial challenges they have endured [7, 21].

In our second theme, *Strategies for Success*, postdocs provide recommendations for maximizing the potential for success in achieving an academic research career. The postdocs overwhelmingly emphasize the importance of possessing a strong love and passion for science. The current literature does not capture this emphasis on passion for success. 'Passion' being the most frequently cited code in this paper, referenced 189 times, highlights the weight of intrinsic values in the supposed impersonal scientific world. As previously mentioned, one respondent shared that "this field is sustained only by passion now." For many, passion is absolutely key for longevity in the field. A unique category that emerged under this theme of passion was characterized by a caution in letting happiness or self-worth depend on scientific research or obtaining an academic position. One postdoc noted: "Don't make your happiness depend on your academic research career." Similarly, another noted: "Do not base your sense of self-worth on having an academic position."

Under our second major theme, the postdocs also expressed frustration about several other aspects of the field. A portion of them adamantly discouraged pursuing academia. Their responses confirm previous reports showing that the current state of academia is underscored by hypercompetitive climates, poor sense of financial security, and a perceived shrink in

available tenure-track faculty positions [22, 25, 42]. One postdoc stated, "Put it this way, even if you are in a top Ivy League school you still need a mentor who will fight for you (I see my supervisor who is an MD/PhD every 4 weeks or so, left to troubleshoot alone), a lab that is well published in your field of study, multiple other postdocs all working in a synergistic way and all the while accepting you live on a 'maybe' with regards to your future and getting paid poorly for it." The perception of most disgruntled postdocs is this: the chances of securing a tenure-track faculty position are slim, even with tremendous passion and sacrifice, largely attributed to conditions that cannot be controlled. Many feel their hard work does not in fact pay off. The alternative perspective, and advice that we summarized from non-disgruntled postdocs: "I understand the chances of obtaining a faculty position and the challenges that come with it. However, I have a passion for this path, and I want to try to achieve it anyway, with the understanding that I will explore alternative career paths if this one does not prove successful." We find this latter frame of mind, coupled with a framework where academic careers are one of many (not better or worse) successful career options for postdoctoral scholars, key for researchers in training.

While we did not ask the postdoc mentors about their trainee attitudes, a recent publication by seven institutions that hold Broadening Experiences in Scientific Training (BEST) programs report the results of faculty surveys of their BEST mentors to understand faculty perceptions around career development for their trainees [43]. The faculty believed that there was a shortage of tenure-track positions and felt a sense of urgency in introducing broad career activities for their trainees [43]. However, many do not feel they have the knowledge and resources necessary to guide and support the 85% of trainees who need professional development for careers outside of academia [22, 41]. The study also found that faculty perceived trainees themselves as lacking in the knowledge base of skills that are of interest to non-academic employers. For budding scientists along the training continuum, the advice given by postdocs in this manuscript could help enhance the knowledge base to best prepare them for non-academic career tracks.

To better illustrate the cost-benefit ratio of pursuing an academic research career, we separated codes into either a benefits or costs category and indicated the frequency of references (Fig 2). Codes that were more associated with the benefits of pursuing an academic research career included academic freedom, strong mentorship, network, and passion, whereas administrative obligations, backup plan, financial stability, and hard work were associated with the costs. While the frequency of codes associated with the costs outnumbered the benefits, 546 to 489, sacrifices (or costs) and benefits are weighed differently between postdoc respondents.

Our study has several limitations. One is that the free-text method of gathering these data prevents engagement with the participant and clarification of intent. With this format, respondents have the ability to address a broad array of topics which promotes variety in their responses but limits our control over what the subjects choose to discuss. Focus groups with postdoctoral participants can explore in more depth the benefits and challenges perceived by postdocs. Due to the subjective nature of the survey respondents' experiences, much of this work also reveals the postdoctoral trainees' perceptions rather than absolute truths or facts about the scientific and career development process. Another limitation is that this sample is not random, and therefore frequency counts should be considered in this context. Given the large number of responses, the frequency by which codes appear can be helpful in understanding trends and emphasis in the sample, but not for comparing significance across codes. In addition, the sample population consists of postdoctoral scholars from top-ranked US universities and institutions, so their perspectives reflect the experience of those who train in similar environments. However, it is important to note that postdoctoral appointees at the top 100 institutions in the country (n = 56,092) account for approximately 88% of the total number of

| Benefits | 489 | 546 | Costs |
|---|---|---|---|
| List | Frequency | Frequency | List |
| academic freedom | 15 | 7 | administrative obligations |
| collaborate | 21 | 29 | backup plan |
| community | 17 | 23 | be comfortable with failure |
| connections | 13 | 11 | be comfortable with uncertainty |
| different academic tracks | 2 | 1 | be ready for difficult relationships |
| enjoy the journey | 13 | 21 | be ready for setbacks |
| explore your capabilities | 2 | 8 | broken system |
| find your own niche | 4 | 1 | burn out |
| if dream, it's worth it | 22 | 15 | demanding work load |
| institutional support | 7 | 15 | family |
| leadership | 3 | 26 | field is competitive |
| learn new skills | 1 | 48 | financial stability |
| long-term gratification | 8 | 79 | hard work |
| mentorship compatibility | 16 | 8 | long hours |
| network | 35 | 41 | long-term commitment |
| passion | 189 | 30 | luck |
| personal brand | 3 | 5 | mental health |
| rewarding | 25 | 48 | need for funding |
| strengthening research skills | 12 | 15 | politics |
| strengthen writing skills | 4 | 37 | sacrifice |
| strong mentorship | 58 | 19 | saturated field |
| teach | 4 | 27 | success not guaranteed |
| transferable skills | 15 | 32 | work-life balance |

**Fig 2. Cost-benefit ratio of pursuing an academic research career.** The codes are listed alphabetically in their respective columns. An end limit of "45" (a double of the average frequency value of 22.5) was set for the data bars to provide a representative visual comparison of the codes' frequencies.

postdocs in the United States (n = 63,861) [38]. Therefore, our sample is representative of a significant portion of the U.S. postdoc population.

Overall, our study shows that most postdocs understand the travails and risks associated with pursuing a tenure-track faculty position in academia. Many perceive the challenges as surmountable and the reward of an academic research career worthwhile. As noted by one postdoc, "Being a successful academic researcher is somewhat akin to pursuing a career in music performance or professional sports. Science and research must be your real passion for which you are willing to work extremely hard and sacrifice. And even with hard work and sacrifice, and of course the requisite level of talent, you may not make it to the big leagues. Be sure you are willing to take this risk and that you can enjoy the journey no matter what happens." These accounts should benefit students and trainees interested in pursuing a research career in academia by helping them make more informed decisions about their career path, ultimately enhancing the scientific workforce.

## Supporting information

**S1 File. Responses of 994 postdoctoral researchers to a single, open-ended survey question: "What advice would you give to someone thinking about an academic research career?".** (DOCX)

**S1 Table. Full list of codes derived from the survey responses.** (DOCX)

**S2 Table. Original codes divided into 6 categories under the major theme, Life in Academia.** (DOCX)

**S3 Table. Original codes divided into 14 categories under the major theme, Strategies for Success.**
(DOCX)

## Acknowledgments

The authors wish to thank Dr. Mary E. Charlson, Dr. Avelino Amado and Dr. Leslie Krushel for helpful comments.

## Author Contributions

**Conceptualization:** W. Marcus Lambert.

**Data curation:** Suwaiba Afonja, W. Marcus Lambert.

**Formal analysis:** Suwaiba Afonja, Damonie G. Salmon, Shadelia I. Quailey, W. Marcus Lambert.

**Funding acquisition:** W. Marcus Lambert.

**Investigation:** W. Marcus Lambert.

**Methodology:** Damonie G. Salmon, W. Marcus Lambert.

**Project administration:** W. Marcus Lambert.

**Resources:** W. Marcus Lambert.

**Software:** W. Marcus Lambert.

**Supervision:** W. Marcus Lambert.

**Validation:** W. Marcus Lambert.

**Visualization:** W. Marcus Lambert.

**Writing – original draft:** Suwaiba Afonja, W. Marcus Lambert.

**Writing – review & editing:** Suwaiba Afonja, W. Marcus Lambert.

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
