## [Decision Letter · Decision Letter 0]

18 Feb 2021

PONE-D-20-40531

Postdocs’ advice on pursuing a research career in academia: A qualitative analysis of free-text survey responses

PLOS ONE

Dear Dr. Lambert,

Thank you for submitting your manuscript to PLOS ONE. After careful consideration, we feel that it has merit but does not fully meet PLOS ONE’s publication criteria as it currently stands. Therefore, we invite you to submit a revised version of the manuscript that addresses the points raised during the review process.

In responding to the reviewers’ questions, please pay particular attention to the following issues:

(1) Your choice and description of the study population.

(2) Concerns raised about your use of grounded theory and the neutrality of nodes and themes you selected.

(3) Comparison of your findings with previous studies to highlight new insights.

We look forward to receiving your revised manuscript.

Kind regards,

Frederick Grinnell

Academic Editor

PLOS ONE

Reviewers' comments:

Reviewer's Responses to Questions

**Comments to the Author**

1. Is the manuscript technically sound, and do the data support the conclusions?

Reviewer #1: Yes

Reviewer #2: Partly

2. Has the statistical analysis been performed appropriately and rigorously? 

Reviewer #1: Yes

Reviewer #2: N/A

3. Have the authors made all data underlying the findings in their manuscript fully available?

Reviewer #1: Yes

Reviewer #2: Yes

4. Is the manuscript presented in an intelligible fashion and written in standard English?

Reviewer #1: Yes

Reviewer #2: Yes

5. Review Comments to the Author

Reviewer #1: This is a timely article that reflects the attitude of postdocs towards pursuing careers in academia in research-intensive institutions as tenure/tenure-track faculty. The open ended free text responses on guidance/advice to graduate students and undergraduates interested in pursuing careers in the biological sciences will certainly be useful as they consider the next steps in their training trajectories.

There are a few questions that would be helpful to address in this article:

1. While I understand that the authors can simply not survey all 21,781 NSF-identified postdocs in the U.S., it is important to explain on what basis the 1,248 survey recipients were selected since this is a very small percentage of total postdocs. What constitutes “top” institutions? On what criteria are those determinations based? How many institutions were included? Are the rankings based on NIH funding, US News and World Report? What is the geographic distribution? How many are public versus private institutions? AAU institutions? Perhaps even naming the institutions would be useful.

2. Were the postdocs primarily in schools or colleges of medicine?

3. What percent of respondents had originally planned to pursue academic careers and how many subsequently changed their minds?

4. How many respondents had any type of individual extramural fellowships (such as the F kiosk from the NIH)

5. Is the use of the IDP mandatory at these institutions? How many respondents actually availed themselves of an IDP to hold discussions regarding career choices with their research mentors?

6. What is the faculty perception among mentors of the postdocs surveyed about their attitudes? Was that even asked in the survey? Even if the question was not asked in the survey, a discussion of the literature around faculty perceptions in the Discussion would be helpful.

Reviewer #2: The manuscript provides a characterization of responses of just under 1000 postdocs to a single question on a survey: “What advice would you give to someone thinking about an academic research career?”. As with any very open question such as this, a great variety of comments were provided. They then used qualitative analysis to characterize the responses. The study provides some new insights, particularly with respect to advice, but not many new insights with respect to perspectives on academic careers beyond what has been quite well documented in previous surveys and studies. The Discussion fails to compare their findings to those of previous studies to highlight new insights. Data on dissatisfaction of postdocs in their current positions and roles does not provide any new insights in particular. At times, the authors come across as having a strong goal of using their data to promote changes in policy with respect to postdoc positions, and in doing so over-interpret the actual words of the respondents. There are several methodological limitations that are not noted, and with such a large sample it seems odd that they did not subdivide responses by one or more demographic categories. If they did, and clearly saw no differences, that should be noted. Detailed comments include:

1. The methodology employed is not grounded theory. Clearly both the author’s previous work and prior studies framed what they expected to see. Thus, it could not be a neutral, outcome-agnostic approach required of grounded theory. It is content or thematic analysis.

2. How the many nodes are combined into the 20 themes they refer to is never presented. Without that, a reader has no idea how many and which of the nodes contributed to the 2 themes they report on here.

3. What were the other themes and why are they not reported on as well?

4. A major limitation of the study is that, as stated in lines 170-171, they only offered the survey to “postdoctoral listservs from top-ranked research universities and institutions”. Thus, by definition, their data to not speak to perceptions and advice of the very many postdocs at other universities. This is a significant limitation that should be noted in the methods as well as in the limitations section so that that readers are fully informed as they consider the data presented. Arguably, the perceptions and advice are contextualized by the extreme research competitiveness and high expectations at these institutions.

5. With such a large fraction of their respondents being international postdocs, it is important that they provide data comparing their responses to non-international postdocs. Evidence of the significance of this population is provided by the average time to the PhD of 4.6 years, which is well-below that for U.S. PhD programs. This likely reflects the shorter, time-constrained PhD training in many countries outside the U.S.

6. It also would be very helpful to know how men and women respond the same or differently. Comparison with UR postdocs would be valuable as well since their numbers are high at 174.

7. It is not clear how the background on postdocs support (intro paragraph 2 and data in results) are germane to the purpose and results of the study. What is presented is true but common knowledge and the results reported don’t add any new insights into this topic.

8. In lines 97-99 the authors state: “This loss of interest may be remedied by an increase in transparency about life in academia and disseminating more information about the critical steps for securing an academic research position [31]”. It is unclear where this statement is coming from as it is counter-intuitive given the assertions leading up to it re: exit from academic careers interests. It would seem that more information about the challenges could only lead to more loss of interest.

9. There was good effort to provide counter-balancing quotations around concerns expressed by any 1 individual with quotations where those concerns were either not seen as challenges or were seen as manageable by others. Any revisions to the MS should continue to provide counter-balancing examples to avoid leading a reader to perceive any single quote as representing ‘truth’ vs. a statement by 1 individual.

10. Results starting on line 292 don’t seem to be germane to this paper and represent the same findings that many other studies have reported previously.

11. Staring on line 316 it appears that statements with respect to professional development support will be forthcoming but those do not materialize. If they want to devote a section to professional development resources it would need to balance those who feel they have adequate support and those who do not. However, this highlights one of the SIGNIFICANT limitations of a single item survey study – with such a broad array of what someone might choose to comment on, there is no way to draw conclusions on importance of peripheral topics such as this. Analytically, one is limited by what the question triggers a respondent to think about, and, for example, career development support would not be a likely topic of primary focus.

12. The results on suggestions for those considering academic careers is by far the most important and useful contribution of this study. It is generally well laid out and could be valuable for PhD students and postdocs. Revisions to the MS should keep that in mind so its centrality is not lost.

13. The Discussion does not significantly speak to how the findings of this study compare to those of previous studies, and what is new about their findings.

14. In lines 611-612 the authors state: “We also suspect that not everyone considers the financial impact of years of training until they are immersed in it.” This seems to be conjecture and no data supporting it come from the study. The follow-on sentence about postdoc frustrations is certainly not new information.

15. Starting on line 631, the paragraph speaks to frustrations with the job market for academic careers, which again is not new information. The authors go on to say, however: “Their responses seem to signal a shift in academia that is underscored by hypercompetitive climates, poor financial compensation, and a shrink in tenure-track faculty positions.” It is not clear why they say this is ‘evidence’ of a shift which most would say has been occurring over the past decades. Furthermore, it is not clear that the number of tenure track positions is declining. Most reports point to how the number of TT positions is flat and all of the growth in academic positions as been in non-TT positions. If there is evidence to support their conclusions it should be referenced.

16. In lines 638-640, the authors conclude: “The crux of this statement suggests that only under perfect conditions will postdocs make it through the academic pipeline.” This is not a conclusion that can be reasonably drawn from the quote upon which it is based. In particular the setting out of ‘perfect conditions’ is not justified from the data provided.

17. Under study limitations, it is important to point out that while the single very open question allows for a wide range of responses, it means that any responses that are very infrequent and/or tangential to what most choose to comment upon must be considered very cautiously. This is true to such a point that they should not be included as quotes in the MS as representative if they come from minor nodes. Since it is not clear which nodes quotes are drawn from, a reader cannot assess if this is true or not.

6. PLOS authors have the option to publish the peer review history of their article (what does this mean?). If published, this will include your full peer review and any attached files.

Reviewer #1: No

Reviewer #2: No

---

## [Author Response · Author response to Decision Letter 0]

4 Apr 2021

Please see rebuttal letter attached.

---

## [Editor Report · Decision Letter 1]

12 Apr 2021

Postdocs’ advice on pursuing a research career in academia: A qualitative analysis of free-text survey responses

PONE-D-20-40531R1

Dear Dr. Lambert,

We’re pleased to inform you that your manuscript has been judged scientifically suitable for publication and will be formally accepted for publication once it meets all outstanding technical requirements. Your revisions answer the concerns and questions raised by the referees.

Kind regards,

Frederick Grinnell

Academic Editor

PLOS ONE

---

## [Editor Report · Acceptance letter]

23 Apr 2021

PONE-D-20-40531R1 

Postdocs’ advice on pursuing a research career in academia: A qualitative analysis of free-text survey responses 

Dear Dr. Lambert:

I'm pleased to inform you that your manuscript has been deemed suitable for publication in PLOS ONE. Congratulations! Your manuscript is now with our production department. 

Kind regards, 

on behalf of

Dr. Frederick Grinnell 

Academic Editor

PLOS ONE